# Insights into the Antioxidant Mechanism of Newly Synthesized Benzoxazinic Nitrones: In Vitro and In Silico Studies with DPPH Model Radical

**DOI:** 10.3390/antiox10081224

**Published:** 2021-07-29

**Authors:** Stefania Marano, Cristina Minnelli, Lorenzo Ripani, Massimo Marcaccio, Emiliano Laudadio, Giovanna Mobbili, Adolfo Amici, Tatiana Armeni, Pierluigi Stipa

**Affiliations:** 1Dipartimento di Scienze e Ingegneria della Materia, dell’Ambiente ed Urbanistica (SIMAU), Università Politecnica delle Marche, via Brecce Bianche, 60131 Ancona, Italy; s.marano@univpm.it (S.M.); e.laudadio@staff.univpm.it (E.L.); 2Dipartimento di Scienze della Vita e dell’Ambiente (DISVA), Università Politecnica delle Marche, via Brecce Bianche, 60131 Ancona, Italy; c.minnelli@univpm.it (C.M.); g.mobbili@staff.univpm.it (G.M.); 3Dipartimento di Chimica, Università di Bologna, via Selmi 2, 40126 Bologna, Italy; lorenzo.ripani2@unibo.it (L.R.); massimo.marcaccio@unibo.it (M.M.); 4Dipartimento di Scienze Cliniche Specialistiche ed Odontostomatologiche-Sez. Biochimica, Biologia e Fisica, Università Politecnica delle Marche, 60131 Ancona, Italy; a.amici@staff.univpm.it (A.A.); t.armeni@univpm.it (T.A.)

**Keywords:** free radicals, nitrone, DPPH, antioxidant, electron paramagnetic resonance (EPR), density functional theory (DTF), cyclic voltammetry (CV), hydrogen atom transfer (HAT), single electron transfer (SET)

## Abstract

Synthetic nitrone spin-traps are being explored as therapeutic agents for the treatment of a wide range of oxidative stress-related pathologies, including but not limited to stroke, cancer, cardiovascular, and neurodegenerative diseases. In this context, increasing efforts are currently being made to the design and synthesis of new nitrone-based compounds with enhanced efficacy. The most researched nitrones are surely the ones related to α-phenyl-tert-butylnitrone (PBN) and 5,5-dimethyl-1-pyrroline N-oxide (DMPO) derivatives, which have shown to possess potent biological activity in many experimental animal models. However, more recently, nitrones with a benzoxazinic structure (3-aryl-2H-benzo[1,4]oxazin-N-oxides) have been demonstrated to have superior antioxidant activity compared to PBN. In this study, two new benzoxazinic nitrones bearing an electron-withdrawing methoxycarbonyl group on the benzo moiety (in *para* and *meta* positions respect to the nitronyl function) were synthesized. Their in vitro antioxidant activity was evaluated by two cellular-based assays (inhibition of AAPH-induced human erythrocyte hemolysis and cell death in human retinal pigmented epithelium (ARPE-19) cells) and a chemical approach by means of the α,α-diphenyl-β-picrylhydrazyl (DPPH) scavenging assay, using both electron paramagnetic resonance (EPR) spectroscopy and UV spectrophotometry. A computational approach was also used to investigate their potential primary mechanism of antioxidant action, as well as to rationalize the effect of functionalization on the nitrones reactivity toward DPPH, chosen as model radical in this study. Further insights were also gathered by exploring the nitrone electrochemical properties via cyclic voltammetry and by studying their kinetic behavior by means of EPR spectroscopy. Results showed that the introduction of an electron-withdrawing group in the phenyl moiety in the *para* position significantly increased the antioxidant capacity of benzoxazinic nitrones both in cell and cell-free systems. From the mechanistic point of view, the calculated results closely matched the experimental findings, strongly suggesting that the H-atom transfer (HAT) is likely to be the primary mechanism in the DPPH quenching.

## 1. Introduction

The ability of nitrones to act as good radical scavengers has been widely investigated over the past decades [1,2,3,4,5]. For years, nitrones have been used as an efficient analytical tool for the detection and characterization of free radicals using EPR spin trapping technique, based on their ability to quickly trap short-lived free radicals and afford persistent and paramagnetic aminoxyl spin adducts [6,7,8]. Due to this unique ability, nitrones have attracted considerable attention as potential therapeutic agents for the treatment of many pathologies where reactive oxygen species (ROS) are implicated, such as cardiovascular diseases, cancer, aging, cerebral ischemia, and various neurodegenerative diseases [9,10,11,12]. Among all nitrone families, two classes have been mainly researched, the cyclic ones derived from the 5,5-dimethyl-1-pyrroline-N-oxide (DMPO) [13,14,15,16,17], and the linear ones derived from the α-phenyl-N-tert-butyl nitrone (PBN) [18,19,20]. Several derivatives of PBN, and to a lesser extent those of DMPO, have been recently synthesized with appropriate chemical modifications that have significantly improved the reactivity of the nitronyl group and the stability of the nitroxide spin-adduct [21,22,23,24]. Although it would be reasonable to attribute their biological action to their spin trapping activity, a clear picture of the actual mechanism of action of nitrones has not yet been reported. On the contrary, a large number of investigations has pointed out that more than one mechanisms may be involved and the free radical trapping activity may be not so likely to be their primary action [24,25,26,27]. For example, there are strong evidences that PBN can inhibit transcription of several pro-inflammatory cytokines in endotoxin-treated rats [28,29,30].

As part of our ongoing research on the reactivity of benzoxazinic nitrones, a series of 3-aryl-2H-benzo [1,4]oxazin-N-oxides have been previously synthesized [31,32], and their reactivity toward several carbon- and oxygen-centered radicals widely explored [27,33]. Results revealed that they act as efficient spin traps for a wide range of free radicals and their activity has been demonstrated to be higher than the commercially available N-tert-butyl-α-phenylnitrone (PBN) [27].

Over the past years, several *para*-substituted nitrones [22,24,34], bearing electron-withdrawing groups have shown improved radical detection and antioxidant properties. The improved reactivity has been ascribed to the electronic nature of the substituent that appears to affect the rate of radical trapping on the nitronyl function by increasing the reactivity toward radical addition reactions [23,34,35]. Therefore, in this study we synthesized two constitutional isomers of benzoxazinic nitrones bearing a methoxycarbonyl group on the benzo moiety (in *para*- and *meta*-position) with the aim to enhance the reactivity of the already studied unsubstituted parent compound. We tested their in vitro antioxidant activity using both cell and cell-free assays. In addition, to gain further information concerning the antioxidant mechanism, their electrochemical properties have been evaluated. All the experimental results were found in good agreement with those obtained from proper density functional theory (DFT) calculations.

## 2. Materials and Methods

### 2.1. Materials and Characterization

α-diphenyl—picrylhydrazyl (DPPH) and 2,2-azobis (2-amidinopropane hydrochloride) (AAPH) were obtained from Merck Co. (Stenheim, Germany). All the solutions were prepared in ultrapure MilliQ water to prevent metal contamination. All the other reagents and chemicals were of analytical grade for biochemical purposes or HPLC grade. All cell culture reagents were purchased from Euroclone, Italy. ARPE19 cells (CRL 2302) were obtained from American Type Culture Collection (ATCC) Manassas, Virginia, USA. Electrochemistry: acetonitrile (MeCN, analytical grade purity over molecular sieves, from Sigma-Aldrich, Germany) and tetrabutylammonium hexafluorophosphate (TBAH, electrochemical and analytical grade from Sigma-Aldrich, Switzerland) were used as received as solvent and supporting electrolyte, respectively.

Electrospray ionization mass spectra were recorded on a Finnigan Navigator LC/MS single-quadrupole mass spectrometer in EI^+^ mode by direct injection of methanol sample solutions. ^1^H and ^13^C NMR spectra were recorded on a Varian Gemini 400 spectrometer in CDCl_3_ at 400 and 100 MHz, respectively. Chemical shifts are reported in ppm relative to residual solvent signals (δ = 7.24 and 77.30 ppm for ^1^H and ^13^C NMR, respectively), and coupling constants (J) in Hz. FT-IR spectra were collected on a PerkinElmer Spectrum GX FT-IR spectrophotometer equipped with ATR single reflection diamond. Measurements were performed with a resolution of 2 cm^−1^, 16 scans over 4000–500 cm^−1^ range. Samples were directly placed on the measuring surface without requiring any preparation. Background adsorption spectrum was recorded each time for correction. Spectra were analyzed on Spectrum 5.3.1 (Perkin-Elmer) operating software. Isotropic X-band EPR spectra were recorded on a Bruker EMX/Xenon spectrometer system equipped with a microwave frequency counter and an NMR Gauss meter for field calibration; for g-factor determination, the whole system was standardized with a sample of perylene radical cation in concentrated sulfuric acid (g-factor = 2.00258).

### 2.2. Synthetic Procedure

The two constitutional isomers of benzoxazinic nitrones (3-aryl-2H-benzo[1,4]oxazin-N-oxides) (2–3) bearing a methoxycarbonyl group on the benzo moiety and the unsubstituted derivative (1) were prepared with a slight modification of the previously reported two-step method [31,32], as illustrated in Scheme 1: alkaline condensation between the appropriate 2-nitrophenol and 2-bromoacetophenone, followed by reductive cyclization (Zn/NH_4_Cl).

6-(methoxycarbonyl)-3-phenyl-2H-benzo[b][1,4]oxazine 4-oxide (2): ^1^H NMR (400 MHz, CDCl_3_) δ (ppm): 3.91 (s, 3H, 6-COOCH_3_), 5.38 (s, 2H, 2-CH_2_), 7.06 (d, J = 14.08 Hz, 1H, arom), 7.26 (CDCl_3_), 7.48 (m, 3H, arom), 8.06 (dd, J_1_ = 10.88 Hz, J_2_ = 1.96 Hz, 1H, arom), 8.24 (dd, J = 08.04 Hz, J = 1.88 Hz, 2H, arom), 8.85 (d, J = 1.92 Hz, 1H, arom). ^13^C NMR (100 MHz, CDCl_3_) δ (ppm): 66.56, 76.77, 76.99, 77.31 (CDCl_3_), 115.99, 121.04, 122.64, 128.06, 128.51, 130.68, 131.17, 149.55. Mp: 118–120. IR (KBr, cm^-1^): 1714 (*v*(C = O) ester), 1435 (δ(CH_3_) methoxy), 1618 (*v*(C = N)), 1266 (*v*(N-O) N-oxide). ESI-MS (m/z) of C_16_H_13_NO_4_: Mr = 283.28, found 283.32. 

7-(methoxycarbonyl)-3-phenyl-2H-benzo[b][1,4]oxazine 4-oxide (3): ^1^H NMR (400 MHz, CDCl_3_) δ (ppm): 3.94 (s, 3H, 7-COOCH_3_), 5.37 (s, 2H, 2-CH_2_), 7.26 (CDCl_3_), 7.48 (m, 3H, arom), 7.66 (d, J = 1.56 Hz, 1H, arom), 7.80 (dd, J_1_ = 8.44 Hz, J_2_ = 1.44 Hz, arom), 8.23 (d, J = 8.36 Hz, 3H, arom). ^13^C NMR (100 MHz, CDCl_3_) δ (ppm): 66.67, 76.67, 76.98, 77.30 (CDCl_3_), 117.44, 121.17, 123.79, 128.13, 131.19, 149.27. Mp: 124–125. IR (KBr, cm^−1^): 1713 (*v*(C=O) ester), 1437 (δ(CH_3_) methoxy), 1608 *v*(C=N), 1265 (*v*(N-O) N-oxide). ESI-MS (m/z) of C_16_H_13_NO_4_: Mr = 283.28, found 283.33.

### 2.3. Determination of Erythrocyte Hemolysis Induced by AAPH

Erythrocyte oxidative hemolysis induced by AAPH and its inhibition in the presence of nitrones was performed as previously described [36]. Blood (5–10 mL) was obtained from healthy consenting donors by venipuncture and collected into tubes containing EDTA as anticoagulant. Total of 5 mL of blood was washed three times in PBS (pH 7.4) by centrifugation at 1000 g for 10 min. The resulting pellet was re-suspended with six-fold PBS (pH 7.4). About 0.5 mL of erythrocyte suspension was added to 0.25 mL of PBS nitrone solutions obtaining a final antioxidant concentration of 0.012, 0.5, and 0.2 mM. The mixtures were incubated for 1 h at 37 °C and then, 0.25 mL of PBS was added with or without AAPH (final concentration AAPH, 50 mM) followed by incubation for 12 h at 37 °C in a gently shaking incubator. 

100% lysis was determined by adding 0.15 mL of a 1% Triton X-100 solution while only PBS was added in the negative control. After sedimentation of the unlysed erythrocytes by centrifugation (1000× *g*, 15 min, 4 °C), the supernatant was transferred to a microtiter plate and diluted 1:4. The hemoglobin absorption was determined at 540 nm using the Synergy HT microplate reader spectrophotometer (BioTek, Winooski, VT, USA). The percent hemolysis was determined using the following Equation (1):%Hemolysis = [(A − A_0_)/(A_total_ − A_0_)] × 100(1)
where A is the absorbance of the test well, A_0_ is the absorbance of the negative control, and A_total_ is the absorbance of the positive control. The mean values of three replicates were reported.

### 2.4. Cell Treatment

ARPE19 cells were maintained in 25 cm^2^ flasks in complete DMEM/F12 medium at 37 °C, 5% CO_2_, and 95% relative humidity. Complete DMEM/F12 medium was prepared by adding 10% (*v/v*) fetal bovine serum (FBS), 2 mM glutamine, and 100 U/mL penicillin-streptomycin. Culture medium was changed every 2 days until cells grew to 90% confluence. The cell cultures were detached by trypsinization with 0.5% trypsin in PBS containing 0.025% EDTA and counted using trypan blue exclusion assay.

For treatments, ARPE19 cells were seeded in 96-well plates at 5 × 10^4^/well to reach 50–60% of confluence at 24 h. In the cytotoxicity assay, the cells were incubated for 48 h with increasing concentrations of nitrones solutions in methanol (3–200 μM). For the cytoprotection assay, the cells were treated with selected nitrones concentrations (6, 12, and 25 μM) and incubated for 24 h; then, cells were treated with AAPH (15 mM) for 6 h. Cellular viability was measured by 3-(4,5-dimethylthiazol-2-yl)-2,5-diphenyltetrazolium bromide (MTT) assay. At the time of analysis, the medium from each well was removed and replaced with fresh medium containing MTT (2 mg mL) and cells were incubated for 4 h at 37 °C in 5% CO_2_ atmosphere. Then, 0.1 mL of DMSO was added to each well until MTT formazan crystals were solubilized. Absorbance was read on a multiwell scanning microplate reader (BioTek Synergy HT MicroPlate Reader Spectrophotometer) at 570 nm using the extraction buffer as blank. The relative cell viability (%) was calculated as (A570 of treated samples/A570 of untreated samples) × 100. Each experiment was performed at least three times in triplicate.

### 2.5. DPPH EPR Signal Quenching Kinetic Behavior

The antioxidant ability of nitrones was evaluated by means of EPR spectroscopy. The decay in the DPPH EPR signal intensity occurs when its EPR-silent diamagnetic counterparts are formed; the test is therefore representative of the nitrones’ ability to react with stable organic free radicals. All the experiments were carried out and at constant temperature of 23 °C in the dark to exclude the possibility of incoming of light-induced effects. EPR spectra were recorded on a Bruker EMX spectrometer at 100 KHz, by following the first line DPPH signal intensity with time at a fixed magnetic field value. In detail, nitrones’ benzenic solutions, prepared in two different nitrone: DPPH molar ratios (1:2 and 1:4) (total volume of 200 µL), were added to a benzenic solution containing DPPH at constant concentration of 800 µM, previously placed within the EPR cavity. The kinetic profiles were obtained following the decay in the EPR signal intensity as a function of time (7200 s) using the 1D_TimeSweep spectra collection mode spectrometer routine. This allowed to obtain the corresponding apparent pseudo first-order reaction rates, determined by regression equations with R^2^ > 0.99. Reaction times used for the calculations typically varied from 30 to 50 s. Measurements were run at least in triplicate.

### 2.6. DPPH Spectrophotometric Assay

The free radical scavenging activity of nitrones was also determined spectrophotometrically using a previously described method [37,38], with minor modifications. For the spectrophotometric evaluations, nitrones and DPPH solutions were mixed (antioxidant 50 µM, DPPH 100 µM, FC) in two different solvents (methanol and acetonitrile), shaken vigorously and incubated for 30 min and 60 min in the dark. The absorbance at 517 nm was determined against a blank which lacked in DPPH.

The percent inhibition of the DPPH radical by antioxidants was calculated according to the following Equation (2):Inhibition ratio (%) = [(A_control_ − A_sample_)/A_control_] × 100(2)
where A_control_ is the absorbance of the control obtained by adding into the DPPH methanol solution a methanol aliquot equal to the antioxidant solution volume. A_sample_ is the absorbance of the reaction solution at 30 min. Each sample was measured in triplicate. Mean and standard deviation (*n* = 3) were calculated.

### 2.7. DFT Calculations

Density functional theory (DFT) calculations were carried out using the GAUSSIAN 09 suite of programs [39], taking advantage of the resources available at Cineca Supercomputing Center [40]. All calculations on paramagnetic species were carried out with the unrestricted formalism, giving S^2^ = 0.7501 ± 0.0003 for spin contamination (after annihilation). Thermodynamic quantities were computed at 298 K by means of frequency calculations performed employing the M06–2X functional in conjunction with the 6-31 + G (d,p) basis set, starting from molecular geometries computed at the B3-LYP/6-31G (d) level of theory. In frequency calculations, negative values (imaginary frequencies) have never been found, demonstrating that all quantities were referred to geometry minima. All calculations were run in vacuo or in acetonitrile (MeCN) by means of the CPCM polarizable conductor calculation model [41,42].

### 2.8. Electrochemical Measurements

Cyclic voltammetry (CV) experiments were carried out in an airtight single-compartment electrochemical cell described elsewhere [43], by using platinum (Pt) as the working and counter electrode and a silver spiral as a quasi-reference electrode. The cell containing the supporting electrolyte and the electroactive compound was dried under vacuum at room temperature for at least 1 h. MeCN was introduced under Argon (Ar). The solution was degassed by performing vacuum/Ar cycles 3-times and afterward left under solvent vapor pressure. All the redox potential (E) values are referred to an aqueous saturated calomel electrode (SCE, which is +0.242V vs. NHE) and they have been determined by adding, at the end of each experiment, ferrocene (Sigma-Aldrich) as an internal standard. The E_½_ potentials have been directly obtained from CV curves as averages of the cathodic and anodic peak potentials for one-electron peaks. The potentials thus obtained were not corrected for (i) the liquid junction potential between the organic phase and the aqueous SCE solution and (ii) the ohmic drop due to the uncompensated resistance between working and reference electrodes.

Voltammograms were recorded either with a potentiostat CH instrument (model 910a) or a custom made fast potentiostat [44] controlled by an AMEL model 568 programmable function generator. The potentiostat was interfaced to a Nicolet model 3091 digital oscilloscope, and the data were transferred to a personal computer by the program Antigona [45].

### 2.9. Statistical Analysis

Data are presented as mean ± S.D. Statistical comparison of differences between antioxidants tested was carried out using Student’s *t*-test. Values of *p* < 0.05 were considered statistically significant.

## 3. Results and Discussion

### 3.1. Synthesis and Structural Characterization

The unsubstituted nitrone, 3-phenyl-2H-benzo[1,4]oxazine 4-oxide, is a known compound, thus, both the details of its synthesis and a full structural characterization can be found in the literature [31,32]. In this study two derivatives of this compound bearing an electron-withdrawing methoxycarbonyl group on the benzo moiety were synthetized and their structure confirmed by mass spectrometry (MS), infrared spectroscopy (IR), and nuclear magnetic resonance spectroscopy (^1^H-NMR and ^13^C-NMR) (see Section 2.2).

### 3.2. Cellular-Based Assays

The effectiveness of nitrones to act as antioxidant molecules was studied in an ex vivo and in vitro cellular models: inhibition of hemolysis of the human erythrocyte and protection from oxidative stress-induced cell death in human retinal cells (ARPE19).

The hemolysis of erythrocytes has been extensively used as an ex vivo model in the study of oxidative stress-induced alteration of cell membranes [46,47,48]. It is an operationally simple and convenient cell-based assay, whereby the AAPH-derived peroxyl radicals generated by thermal decomposition induce lipid peroxidation and oxidation of membrane proteins. This eventually leads to the formation of holes in the membrane and ultimately hemolysis. The use of the free radical reactions initiator AAPH is recommended as more appropriate to measure radical-scavenging activity in vitro as the activity of peroxyl radicals shows a greater similarity to cellular activities such as lipid peroxidation [49]. Based on the nature of the radicals and the cell model used, this method is biologically relevant and appropriate for estimating the antioxidant activity of novel compounds.

First, we assessed the toxic profile of nitrones on erythrocytes (RBCs). As shown in Figure 1A, concentrations higher than 100 µM induce a certain degree of hemolysis with respect to untreated RBCs (* *p* < 0.001). Since cytotoxicity is indicated at over 10% hemolysis [50], we used 12, 25, and 50 µM concentrations to determine the ability of nitrones to counteract hemolysis induced by oxidative stress.

Erythrocytes were therefore pre-treated with nitrones at three concentrations of 50, 25, and 12 µM, followed by the addition of AAPH. Figure 1B shows the percentage of survival of the erythrocyte population, which is directly related to the protective effect of nitrones. It can be clearly seen that, among all the nitrones tested, nitrone 3, bearing the methoxycarbonyl group in *para* position, showed the highest ability to protect the erythrocyte cells from AAPH-induced hemolysis at all concentrations.

However, as the concentration decreases, nitrone 2 reached its highest protection effect at 50 µM, which is statistically greater than that observed for nitrone 1. Instead, at 25 µM there are no differences in their effect and both are able to inhibit hemolysis of RBCs. On the other hand, nitrone 1 shows its best performance at the lowest concentration tested (12 µM), overcoming the protection effect of nitrone 2. Experiments performed on human retinal pigment epithelial cells (ARPE19) show similar results (Figure 1C,D): at the lowest concentration tested, all nitrones are able to protect the cells from oxidative stress-induced cell death and the efficacy follows the same trend observed in erythrocytes (nitrone 3 > nitrone 2 > nitrone 1). However, for nitrone 1, as the concentration increases there is a loss of cellular protection and the percentage of cell viability is comparable to that of AAPH-treated cells. In contrast, nitrone 2 and nitrone 3 are effective also at 12 µM concentration inducing 10%-increase in cell viability compared to untreated cells (*p* < 0.05) (see Figure 1D). Overall, in all cases, nitrone 3 displays the greatest ability to protect erythrocytes and ARPE19 cells from the oxidative stress-induced damage at all concentrations tested.

### 3.3. DPPH EPR Signal Quenching

The DPPH radical, having an unpaired electron, generates a characteristic and well-known five-line EPR spectrum [51], whose intensity is proportional to DPPH concentration, and its decay with time in the presence of antioxidants denotes a DPPH radical transformation into the corresponding diamagnetic derivatives. Figure 2A–D shows the EPR kinetic profiles for the reaction between nitrones and DPPH in the two different molar ratios (1:2 and 1:4). As can be clearly seen from the corresponding traces, *para*- and *meta*-substituted nitrones (A–B) are able to quench DPPH and the extent of the signal reduction is dose-dependent, achieving up to approximately 37% and 15% decrease in the DPPH activity at the highest concentrations (1:2 nitrone:DPPH molar ratio), respectively. For nitrone 1, it is interesting to note (Figure 2D) that in a previous experiment the unsubstituted nitrone 1 showed no activity toward DPPH and its addition led to an initial growth of the DPPH signal, up to ~6% for the 1:2 molar ratio. Such a finding implies that freshly prepared DPPH solutions might contain a certain fraction of diamagnetic reduced DPPH (DPPH-H), which were rapidly oxidized to DPPH radical by nitrone 1.

In order to verify whether the starting solution of freshly prepared DPPH contains a fraction of DPPH-H, an oxidizing agent (PbO_2_) was added into the DPPH solution and the corresponding EPR signal was followed with time. Figure 2E shows the growth of DPPH signal within the first 10 min after which the equilibrium was reached (100% of DPPH-H conversion into DPPH radical), confirming the presence of DPPH-H in the freshly prepared solution. In light of this observation, the experiments were repeated using PbO_2_-treated DPPH as a starting solution. While no changes in the DPPH decay profiles and percentage of signal reduction were observed for nitrone 2 and 3, nitrone 1 did no longer induce signal growth. Under these conditions, nitrone 1 (C) was found to be able to reduce the DPPH EPR signal intensity, and the extent of reduction was equal to 9.67 and 2.70% for the 1:2 and 1:4 molar ratios, respectively.

Overall, the presence and the position of the electron-withdrawing methoxycarbonyl group appears to increase the reactivity of benzoxazinic nitrones toward DPPH if compared to the unsubstituted parent compound. In particular, in accordance with the protective activity trend observed in AAPH-treated erythrocytes and ARPE19 cells, derivative 3, bearing the substituent in *para* position exhibited the highest activity, followed (in a decreasing order) by the *meta*-substituted isomer and the unsubstituted one.

### 3.4. DPPH Spectrophotometric Assay

In order to see whether the DPPH quenching activity behaviors of nitrones were reproducible and independent by potential solvent-solute interactions, the decay profiles in the absorbance of the reaction mixtures at 1:2 nitrone:DPPH molar ratio were also measured by UV/VIS spectroscopy at 517 nm in a polar protic solvent (methanol-MeOH) e in a polar aprotic solvent (acetonitrile-MeCN). Figure 3 shows the percentage of inhibition of DPPH based on the measured absorbance values taken after 60 min of incubation in acetonitrile (red) and methanol (blue). Except for a slight increase in the reactivity of nitrone 3 in MeCN, it is clear that for all nitrones, the percentages of neutralization of DPPH radical observed by UV/VIS spectroscopy are in good agreement with those observed by EPR in benzene (~10%, 14% and 33–37% for nitrone 1, 2 and 3, respectively). This indicates a good reproducibility of the results from the two methods employed, despite the very different properties of the solvents used (methanol, acetonitrile, and benzene).

### 3.5. Insights into the Mechanism of Antioxidant Activity of Nitrones

Nitrones are known spin traps that are capable of trapping free radicals (carbon-, oxygen- and nitrogen-centered radicals) via addition reactions at the C=*N* double bond of the nitronyl function, thus, encouraging their use as pharmacological agents against oxidative stress-mediated pathologies. However, the mechanism by which nitrones exert their antioxidant activity is still not completely clear. Besides the well-documented radical trapping ability of nitrones, a large body of investigations suggest that more than one mechanism may be involved [24,25,26,27]. In the specific case of DPPH, chosen as model radical, Scheme 2 summarizes all the possible reaction mechanisms that could take place between DPPH and the nitrones under investigation. In a typical DPPH assay, nitrones are therefore expected to quench DPPH free radicals via a spin trapping mechanism following pathway (a), giving rise to a nitroxide spin adduct (compound 4). However, quenching of DPPH radical by either sequential electron transfer (SET) (pathway (b)) and/or hydrogen atom transfer (HAT) (pathway (c)) are also possible. The H-donating ability could be associated to either the potentially removable hydrogens in position 2 (i.e., oxygen α position) or the hydrogen from the N-OH group of the corresponding hydroxylamine tautomer (compound 5); such hydrogen atoms could be abstracted by DPPH radicals affording either radical compound 8 or the nitroxide compound 7, likely followed by addition of another DPPH molecule.

#### 3.5.1. Feasibility of Pathway (a): Radical Trapping Ability of Nitrones

As previously mentioned, the unique chemical structure of nitrones makes them excellent spin traps, able to react with many biological transient free radicals to form persistent paramagnetic nitroxide spin adducts, which can be detected and quantified by EPR.

However, in the specific case of reaction with DPPH model radical, the spin trapping mechanism can be excluded because no EPR signal attributable to the corresponding spin adduct 4 were recorded during the EPR experiments; this could be attributed to unfavorable steric factors of DPPH molecule that may hinder its addition to the nitrone C=N double bond.

#### 3.5.2. Feasibility of Pathway (b) and (c): Single Electron Transfer (SET) and Hydrogen Atom Transfer (HAT)

To explore the feasibility of HAT and SET mechanisms, proper DFT calculations were run in order to estimate the corresponding bond dissociation energies (BDE), electron affinities (EA), and ionization potential (IP).

In fact, in a common single electron transfer reaction, the antioxidant is expected to undergo oxidation and protect the substrate; in general, the lower the IP, the easier is one-electron oxidation, and vice versa for EA. In our case, nitrones should yield the corresponding radical cation 6, while DPPH radical should be transformed in its anion. Therefore, the feasibility of such a mechanism is regulated by the nitrone IP and DPPH EA. However, considering that both IP and EA are quantities referred to the gas phase, for reactions taking place in a solvent the corresponding redox potentials (*vide infra*) also play an important role.

On the other hand, the HAT mechanism foresees the transfer of an H atom from the chain-breaking antioxidant to the radical species involved in the oxidation process [38,52]. In such cases, the BDEs of the involved hydrogen (H) atoms are crucial. As mentioned, the nitrones under investigation have two possible sites of H atom donation, at C(2) and the N-OH group of the corresponding hydroxylamine tautomer 5. However, the latter site could be excluded as in our experiments, an EPR signal attributable to the resulting nitroxide 7 was not detected.

Table 1 summarizes the BDE and IP values (Kcal/mol) computed both in the gas and in solvent phases (MeCN) for all the investigated compounds to obtain suitable thermodynamic parameters and estimate the feasibility of both SET and HAT mechanisms. Above all, the resulting data showed unexpected low BDEs for one H-atom abstraction at C(2) for all compounds, suggesting that these nitrones can act as potential H-donors. In this regards, it is interesting to note that BDE values in the gas phase are always approximately 20 kcal/mol lower that those reported for most well-known phenolic antioxidants (BDE-OH value of about 80 kcal/mol) [53,54]. Moreover, despite the contribution attributable to the presence of the solvent in stabilizing the ionic intermediates foreseen in the SET mechanism, the sign of the corresponding Gibbs Free Energy changes (ΔG) remained positive, indicating a non-spontaneous process. In contrast, the quantities for the HAT process show a negative sign, indicating that this process is thermodynamically favored. These findings are also supported by a proper electrochemical investigation (see following section).

However, since all nitrones are characterized by similar IP, EA, and BDE values, a discrimination among them is not feasible upon this basis. Therefore, the experimentally different activities found among the nitrones may be more likely attributed to kinetic rather than thermodynamic factors (see Section 3.5.5).

#### 3.5.3. Cyclic Voltammetry

The electrochemical investigation of all derivatives involved in this study was carried out in acetonitrile (MeCN) by cyclic voltammetry (CV) at room temperature, using platinum disk as working electrode. Redox potentials of the various processes are collected in Table 2 and referred to saturated calomel electrode (SCE). The cyclic voltametric curves of the three nitrone derivatives and DPPH are reported in Figure 4A–D, respectively.

The voltametric curves of all nitrones herein studied (Figure 4A–C)) are characterized by electron transfer affected by chemical irreversibility, both for the reduction and oxidation processes. Such an irreversibility was also checked performing the investigation at higher scan rate (i.e., up to 20 V/s) and lower temperature (i.e., −40 °C), evidencing a fast reaction following up the electron transfer (data not reported and discussed herein). The (unsubstituted) nitrone 1 voltammogram (A) shows a one-electron oxidation at a peak potential (Ep) of +1.56 V and a one-electron reduction with an Ep = −1.84 V. The *meta*-substituted nitrone 2 voltammogram (B) shows two oxidation processes at Ep = +1.72 V and +2.03 V. At negative potentials, this derivative shows a mono-electronic reduction at −1.59 V (Ep) and a subsequent reduction peak attributed to the reduction of the resulting reaction product formed. Lastly, the *para*-substituted nitrone 3 shows a voltametric behavior (C) similar to that of the *meta* analogue 2, with two oxidation peaks at Ep of +1.68 V and +2.00 V, a reduction at −1.52 V (Ep), and a subsequent further reduction of the product formed in the first reduction step. From these results, it appears that the introduction of a methyl-acetate substituent on the nitrone benzo moiety, acting as electron-withdrawing group, leads to an increase of the oxidation potentials of nitrones 2 and 3 with respect to the corresponding oxidation of unsubstituted nitrone 1. The same trend is also observed by DFT-computed IP (ionization potentials) values (in the gas phase) reported in Table 1. 

Moreover, in Figure 4, the cyclic voltammogram of DPPH (D) shows a reversible one-electron oxidation and two reductions processes of the relatively stable DPPH radical, in agreement with previous literature reports [55,56,57], with half-wave potentials (E_1/2_) of +0.76 V, +0.23 V, and −1.30 V, respectively. The voltametric features of the second reduction process reveal a sluggish electron transfer process. Finally, considering all electrochemical parameters determined, we can confidently expect that a SET mechanism with DPPH is thermodynamically unfavorable.

#### 3.5.4. UV/VIS Evidences

An additional experimental evidence, reinforcing the predominance of a HAT over a SET mechanism, can be found in the time courses of UV/VIS experiments, where characteristic spectral changes have been observed upon mixing of DPPH with nitrones in either MeCN or MeOH. In fact, when a one-electron oxidation/reduction takes place, the expected decrease in the absorption band at 517 nm due to DPPH radical should be accompanied by the appearance of an absorption one at either 380 nm, diagnostic of the formation of DPPH cation (DPPH^+^), or at 438 nm in the case of DPPH anion (DPPH^−^) formation [58,59]. Figure 5A,B shows the spectral changes in the reaction mixtures between DPPH and nitrones in MeCN (A) and in MeOH (B) after 30 min of incubation. It can be clearly seen that in the presence of nitrones, neither the absorbance band at 380, nor the band at 438 nm were observed, confirming that the reaction between nitrones and DPPH does not proceed via an electron-transfer process.

#### 3.5.5. EPR Kinetic Analysis

Since all studies carried out indicate that a SET process should not take place between the studied nitrones and DPPH radical, and no significant differences in the thermodynamic parameters among nitrones were found, the different reactivity of nitrones toward DPPH and in both the protection of erythrocytes and ARPE19 cells from APPH-induced oxidative stress could be rather ascribed to kinetic factors governing the thermodynamically feasible HAT process. Therefore, the reaction kinetic behavior between all nitrones and DPPH is herein investigated.

As previously shown in Section 3.3 the reaction between nitrones and DPPH is characterized by a relatively faster decay in the DPPH EPR signal intensity at the beginning, followed by a slower step in which an equilibrium is likely reached. In accordance with previous reports [60,61,62], the reaction initial period is considered the most relevant step for rate constant evaluation, since the reaction is expected to follow a second-order kinetics, thus, being dependent on both DPPH and nitrone concentrations.

To evaluate the reaction rate constant, proper EPR experiments were set to evaluate the EPR DPPH signal decay with time, as described in the corresponding section. Briefly, pseudo-first-order experimental conditions were considered using a 1:2 nitrone:DPPH molar ratio, and the logarithm of the signal intensities recorded in the initial regions of the decay curve were reported over time (Figure 6A). Figure 6B displays an example of the linear fitting of the experimental initial DPPH EPR signal decay in the presence of Nitrone 3 with a fitting of R^2^ > 0.99. Rate constants were obtained by the line slope.

From Figure 6A, it can be clearly seen that nitrone 3 displays the highest apparent rate constant (2.4 × 10^−3^ s^−1^) while no significant differences have been found between nitrone 1 and 2 (~1.7 × 10^−3^ s^−1^). Considering that reactions following an “outer sphere” SET mechanism are typically much faster [63], these results strongly argue in favor of HAT process rather than a SET mechanism, in full agreement with all the data discussed herein, including indications arising from DFT calculations.

## 4. Conclusions

In this study, two new benzoxazinic nitrones were synthesized introducing a methoxycarbonyl group at 6 and 7 positions of the benzo- moiety; the electron withdrawing effects of these substituents on their antioxidant activity was compared with respect to the corresponding unsubstituted analogue. More in details, the activity of all nitrones was evaluated in vitro by their inhibitory effects against both hemolysis in human erythrocyte and cell death in ARPE-19 cells, as well as by means of typical DPPH quenching assay. The results indicate that in all cases the presence and the position of the substituent appears to affect the nitrones’ antioxidant activity. In particular, the derivative substituted in the *para* position with respect to the nitronyl function, exhibited the greatest ability to protect erythrocytes and ARPE19 cells from the oxidative stress-induced damage at all concentrations tested, as well as the highest reactivity toward DPPH radical.

To rationalize the experimental findings and provide insights into the most plausible mechanism of the antioxidant activity of nitrones, quantum chemical DFT calculations were performed. Results suggested that HAT is surely the most thermodynamically favorable antioxidant mechanism of nitrones toward DPPH, both in the gas and in solvent (MeCN) phases. These results indicated an unexpected, good tendency of benzoxazinic nitrones as H atom donors, also strongly supported by further experimental evidence. In particular, cyclic voltammetry confirmed a thermodynamically unfavorable SET mechanism, as well as the time course UV/VIS experiments showed a complete absence of absorbance bands at 380 (diagnostic of DPPH^+^) and 438 nm (DPPH^-^).

Since a decisive discrimination on nitrones’ antioxidant activity was not possible from a thermodynamic point of view, we used a kinetic approach to explain the highest activity of nitrone 3 shown in all cases. In these experiments, carried out by EPR spectroscopy, nitrone 3 was indeed found to react with DPPH with a higher rate compared to nitrones 1 and 2.

However, to gain a more in-depth understanding onto the mechanism of antioxidant action of benzoxazinic nitrones, further investigations are still in progress. To this end, we will be attempting to run those reactions in a larger scale with the aim to isolate and characterize the products eventually arising from nitrones and DPPH.

## Data Availability

All data are contained within the manuscript.

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
