# Peer review of "Insights into the Antioxidant Mechanism of Newly Synthesized Benzoxazinic Nitrones: In Vitro and In Silico Studies with DPPH Model Radical"

_antioxidants, 2021, doi:10.3390/antiox10081224_

Round 1

Reviewer 1 Report

It is well revised based on the comments from reviewers. Therefore, It is acceptable for the publication on Antioxidants its present form. 

Author Response

Thank you very much for your positive feedback.

Reviewer 2 Report

The authors have significantly improved the article compared to the previous version. The paper makes a good impression. However, there are some comments:

  1. It looks a bit illogical that the authors first investigate the antioxidant properties of nitrones, and then investigate their cytotoxicity. As a result, it turned out that the authors investigated the inhibition of hemolysis by nitrones at concentrations at which cells die. The authors should first determine the IC50 value, and then, at this and lower concentrations, determine the antioxidant activity of nitrones and inhibition of hemolysis. Also, the authors should determine the optimal concentration of nitrones for antioxidant protection of cells and, using this concentration, determine at what concentration of AAPH the antioxidant property disappears. In my opinion, this part of the article requires revision.
  2. For synthesized compounds 2 and 3, the authors should provide graphical NMR and HRMS spectra. In ESI-MS data, there is a very large difference between found and calculated values. The difference should not exceed 0.003 m / z. Authors should double-check the data.

After corrections, the article can be considered for publication in Antioxidants.

Author Response

Dear Reviewer, we appreciated your comments.

  1. Section 3.2. and the Figure 1 is now corrected based on your suggestions.
  • The cytotoxicity studies on human retinal cells (ARPE19) showed that all nitrones are cytotoxic at concentrations higher than 25 µM. As you rightly pointed out, we want to determine the antioxidant activity of nitrones and therefore the antioxidant protection experiments on ARPE19 cells were performed at concentrations equal and below 25 µM. In fact, in the presence of cytotoxic effect we could not see any antioxidant protection and therefore, we should not test the antioxidant activity at concentration corresponding to the IC50 values.
  • Regarding the AAPH concentration used, previous MTT viability assays for antioxidant studies on ARPE19 cells were performed to establish the combination of dose/time of AAPH treatment with 15 mM representing the IC50 value after 24 h of treatment. We have already used this cellular model in a previous work (doi: 10.3390/antiox8080258).
  • In the haemolysis inhibition, some nitrone concentrations tested are cytotoxic in ARPE19 cell culture (50 and 200 µM). However, the two cellular models (erythrocyte and cell culture) are different assays, thus, the response is expected to be different as well. In fact, in the haemolysis assay, the AAPH concentration able to induce 50% of haemolysis was about 3 times higher compared to the one used in the cellular experiment. As a proof of that, we also added the toxic profile of nitrones on erythrocyte that clearly shows the use of non-toxic nitrone concentrations.
  1. Please find attached the graphical 1H-NMR spectra of the two derivatives as requested and please see the modified structure assignments (in revision mode in section 2.2). We apologize for the imprecise interpretation reported in the previous version. Regarding the HRMS, we consulted the journal policies and found that there are no specific requirements for reporting High Resolution mass spectra data for Antioxidants. We have repeated the MS experiments to double check the data and reported the new results. We hope that after this careful revision of MS and NMR data, the manuscript now provides adequate structural characterization of our synthesized compounds.

Reviewer 3 Report

In this article Marano et al. habve synthetized two benzoxazinic nitrones and then evaluated their antioxidants activity.
This article is a good work and should be published after minor revisions:
-    In the synthesis the authors reported two different mass for compounds 2 and 3; but these compouds are positional isomers so they should present the same ESI-MS (m/z). For compound 2 the authors reported the Mr value but with the used instrument [Mr+H] value, as reported for compound 3, is the corrected value.
-    In the 1H-NMR spectra of compound 2 the authors reported 9 aromatic protons but the molecule possess only 8 aromatic protons; the authors should correct the NMR spectral data.

Author Response

Dear Reviewer, thank you very much for your valuable comments.

  1. We have repeated the MS experiments to double check the results and we can now confirm that both derivatives are detected as molecular species produced by oxidation rather than protonation, so the mass correspond to M+. This has also been observed for other nitrones and nitroxides (see references:

https://doi.org/10.1002/(SICI)1096-9888(200005)35:5<607::AID-JMS967>3.0.CO;2-7; https://www.sciencedirect.com/science/article/abs/pii/S0040402001884301).

  1. We apologize for the erroneous 1H-NMR structural assignments reported in the previous version. We have modified the related section and reported corresponding graphical spectra for verification.

Round 2

Reviewer 2 Report

The manuscript was revised based on the comments from reviewers. Therefore, It is acceptable for the publication on Antioxidants its present form.